# Near-room-temperature martensitic actuation profited from one-dimensional hybrid perovskite structure

Bei-Dou Liang [1], Chang-Chun Fan [1], Cheng-Dong Liu[1], Chao-Yang Chai[1], Xiang-Bin Han [1] ✉ & Wen Zhang [1] ✉

Martensitic transformation, usually accompanied by ferroelastic and thermo-elastic behaviors, is an interesting and useful mechanical-related property upon external stimuli. For molecular crystals, however, martensitic systems to show reversible stimuli-actuation behaviors are still limited because of a lack of designability and frequent crystal collapse due to large stress releases during the transformations. Here, a one-dimensional hybrid perovskite semi-conductor (NMEA)PbI$_3$ (NMEA = N-methylethylammonium) was prepared by following a dimensionality reduction design principle. The crystal undergoes reversible ferroelastic and thermoelastic martensitic transformations, which are attributed to weak intermolecular interactions among the chains that easily trigger the interchain shearing movement. The actuation behavior occurring during the phase transition is very close to room temperature and demon-strated to behave as a mechanical actuator for switching. This work provides an effective approach to designing molecular actuators with promising applications in next-generation intelligence devices.

Solid-state structural phase transitions have attracted much attention as they can bring about valuable physical/chemical properties and functions like ferroicity[1–6], dielectric transition[7], spin transition[8,9], or dynamic response[10] in crystals. Among them, the dynamic response is a fascinating one that shows mechanical motions upon external stimuli (light, heat, force, electric/magnetic field, etc). The crystals can act as actuators to expand, contract, deform, bend, twist, or burst by con-verting the stimuli into mechanical responses. However, because of a lack of designability, the synthesis of such materials heavily relies on serendipity. Furthermore, most crystalline state transformation pro-cesses accompanied by significant response behaviors are irreversible and even result in severe destruction due to splitting and exploding[11–13]. Additionally, many reversible actuators respond only under extreme conditions, greatly limiting their practical applications. Therefore, it has long been a pursuit to design and synthesize fully reversibly stimuli-responsive materials actuated under ambient conditions.

Martensitic transformation is a special kind of structural phase transition that usually brings about stimuli-responsive phenomena. It is a displacive-type solid-state phase transition with basic character-istics of diffusionlessness, first order, low transition barrier, ultrafast kinetics, cooperative displacement, habit plane, and structural reversibility[14,15]. The concept of martensitic transformation was first proposed in metallurgy. By quenching carbon steel in the austenite phase, martensite occurs with excellent performance of improved strength, hardness, and toughness[16]. Although the realm of martensitic transformation has expanded into molecular crystals in the past few years, the number of material systems is still limited and the trans-formation mechanism needs deep investigation[15,17,18]. The reason for the great interest in molecular martensitic transformations can be attributed to the ultrafast and reversible phase transitions in response to thermal/mechanical stimuli and the transition usually bring about fascinating properties such as ferroelasticity, thermoelasticity, super-elasticity, and shape memory effect[15,19]. These properties are attractive

[1]Jiangsu Key Laboratory for Science and Applications of Molecular Ferroelectrics, School of Chemistry and Chemical Engineering, Southeast University, Nanjing 211189, China. ✉e-mail: hanxiangbin@seu.edu.cn; zhangwen@seu.edu.cn

and play an important role in mechanical applications. It should be mentioned that the ferroelastic phase transition is often thought to be a subgroup of martensitic transformation. Although they share the basic characteristics, the former is characterized by spontaneous strain and a unique strain-stress hysteresis response in the ferroelastic phase which obeys certain symmetry-breaking rules when the phase transition occurs.

To our knowledge, the known martensitic transformations in molecular crystals are largely limited to pure organic materials and coordination compounds. We hope to expand molecular martensites to hybrid halide perovskite materials for their outstanding optical and electrical properties[20–23]. From the material designing perspective, structural dimensionality has a great impact on the martensitic transitions. As a representative of low-dimensional perovskites, one-dimensional (1D) perovskites have superior structural anisotropy features in addition to excellent stability[21,24]. Unlike the isotropic Pb−I bonds in 3D perovskites such as $(MA)PbI_3$ (MA = methylammonium), the 1D perovskite structure only has strong bonds within the Pb−I chain. Due to the blocking of organic cations, the 1D chains are bound together by weak van der Waals interactions. The moderate interactions among the chains and crystallographic anisotropy can ease the interchain movement and further martensitic transformation (Fig. 1). In contrast, 2D and 0D perovskites are less favored structures suffering from either strong interlayer interactions that impede relative slipping-like displacements or undesired electronic structures and poor semiconducting properties. Meanwhile, the transition temperatures of the perovskites, as a key parameter for practical applications, can be easily tuned to room temperature by various methods such as group/component substitution strategy (e.g., halogen and metal), isotope effect, and doping/mixing. These features offer greater possibilities for 1D phase transition perovskites with intriguing properties[25–28]. However, in reported 1D perovskites, researchers have just put focuses on the characterizations of the ferroelastic properties, but ignored their martensitic transformation nature and potential thermoelastic actuation behaviors[29–31].

In this work, we report a 1D hybrid organic-inorganic perovskite $(NMEA)PbI_3$ (1; NMEA = N-methylethylammonium) experiencing two reversible ferro- and thermo-elastic martensitic transformations. Significant shearing behavior around room temperature endows 1 with remarkable mechanical actuation properties. This finding expands the realm of molecular martensitic transformations to 1D halide perovskites by exploiting intrinsic structural anisotropy.

## Results and discussion
### Structural phase transition
Yellow crystals of 1 were grown from a HI solution containing $PbI_2$ and NMEA salt. Phase purity of the crystalline samples was confirmed by powder X-ray diffraction (Supplementary Fig. 1). Thermogravimetric

analysis measurement shows the decomposition begins at about 500 K, indicating good thermal stability (Supplementary Fig. 2). Single crystal orientation is determined for subsequent characterizations (Supplementary Fig. 3).

The phase transitions in 1 were first revealed by differential scanning calorimetry (DSC) measurement (Fig. 2a). Two pairs of reversible thermal peaks appear at 206/219 K and 274/298 K at a ramping rate of 20 K/min, indicating two structural phase transitions. Similarly, temperature-dependent dielectric constant ($\varepsilon = \varepsilon' - i\varepsilon''$) measurement performed on a powdered crystalline sample shows two distinct dielectric anomalies, corresponding well with the DSC curves (Fig. 2b). Upon heating to 223 K, the dielectric constant jumps from 10 to 30, showing a change of three times. The phase transition at room temperature also displays a comparable dielectric change. Such distinct and reversible dielectric transitions at different temperature ranges not only provide insight into the dynamics of dipoles from a mechanistic point of view but also endow 1 potential application in switchable dielectrics[7,32]. Variable-frequency measurement reveals no dispersion behavior at 1–1000 kHz, suggesting the polar components respond quickly to the external electric field (Supplementary Fig. 4).

### Single crystal structure analyses and martensitic phase transition
To figure out the mechanism of phase transition behavior in 1, variable-temperature single-crystal X-ray diffraction was performed at 153, 253, and 333 K (Supplementary Table 1).

In the low-temperature phase (LTP), 1 crystallizes in the lowest symmetry triclinic $P\bar{1}$. Characteristic face-sharing $[PbI_3]_n$ chain running along the [100] forms the inorganic anion skeleton, and the ordered NMEA cations locate around the inorganic anion chains as counterions (Fig. 2c). Atomic contacts to the Hirshfeld surface of the $[PbI_3]_n$ show that much weak interactions exist between the chains and NMEA cations, which further supports the analysis of easy movement between the chains (Supplementary Fig. 5). The Pb−I distances range from 3.0992 to 3.4007 Å (Supplementary Table 2), indicating a severely distorted octahedron configuration.

In the intermediate-temperature phase (ITP) at 253 K, the space group of 1 changes to monoclinic $C2/c$, and the originally ordered cations exhibit a degree of disorder (Supplementary Fig. 6). Since a twofold axis passes through the geometric center of the NMEA cations, carbon and nitrogen atoms in the middle of the cation are indistinguishable (Fig. 2d, Supplementary Fig. 6). The inorganic anion skeleton also increases symmetry with the space group change. The Pb−I distances range from 3.2047 to 3.2394 Å, showing relatively close values to those in the LTP. Synergistic movement of the organic cations and inorganic chains initiates shear and anisotropic change of the unit cell. Along with the transition from the triclinic to monoclinic crystal system, the LTP-ITP transition presents a sharp elongation of the a-axis, while the b-axis shows a contrary tendency to shorten (Fig. 3a). The length changes of the a and b axis are calculated to exceed 8% and −4%, respectively, and the cell angles also display remarkable changes (Fig. 3b).

As to the high-symmetry hexagonal $P6_3/mmc$ in the high-temperature phase (HTP), all the Pb−I distances become the same value of 3.2249 Å, making $[PbI_6]$ a regular octahedral geometry. The NMEA cation adopts a completely disordered state over three positions (Fig. 2e, Supplementary Fig. 6). Such a highly symmetrical structure reflects a delicate balance between the thermal vibration and internal stress in the crystal. For the ITP-HTP transition, a significant cell shearing perpendicular to the c axis is clearly observed from the side view of the inorganic chains and the amplitude of variation for the a and b axis is ±6%.

Discontinuity in the cell parameter vs temperature diagram reveals the abrupt phase transition feature, and the dramatic cell parameter variation suggests a uniqueness of the phase transitions

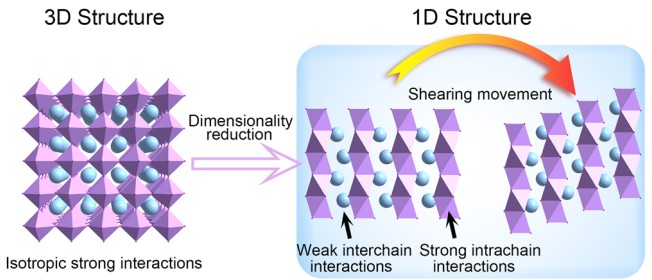

**Fig. 1 | Design strategy for martensitic transformation systems.** Dimensionality reduction strategy for the design of martensitic actuators by facilitating shearing movement.

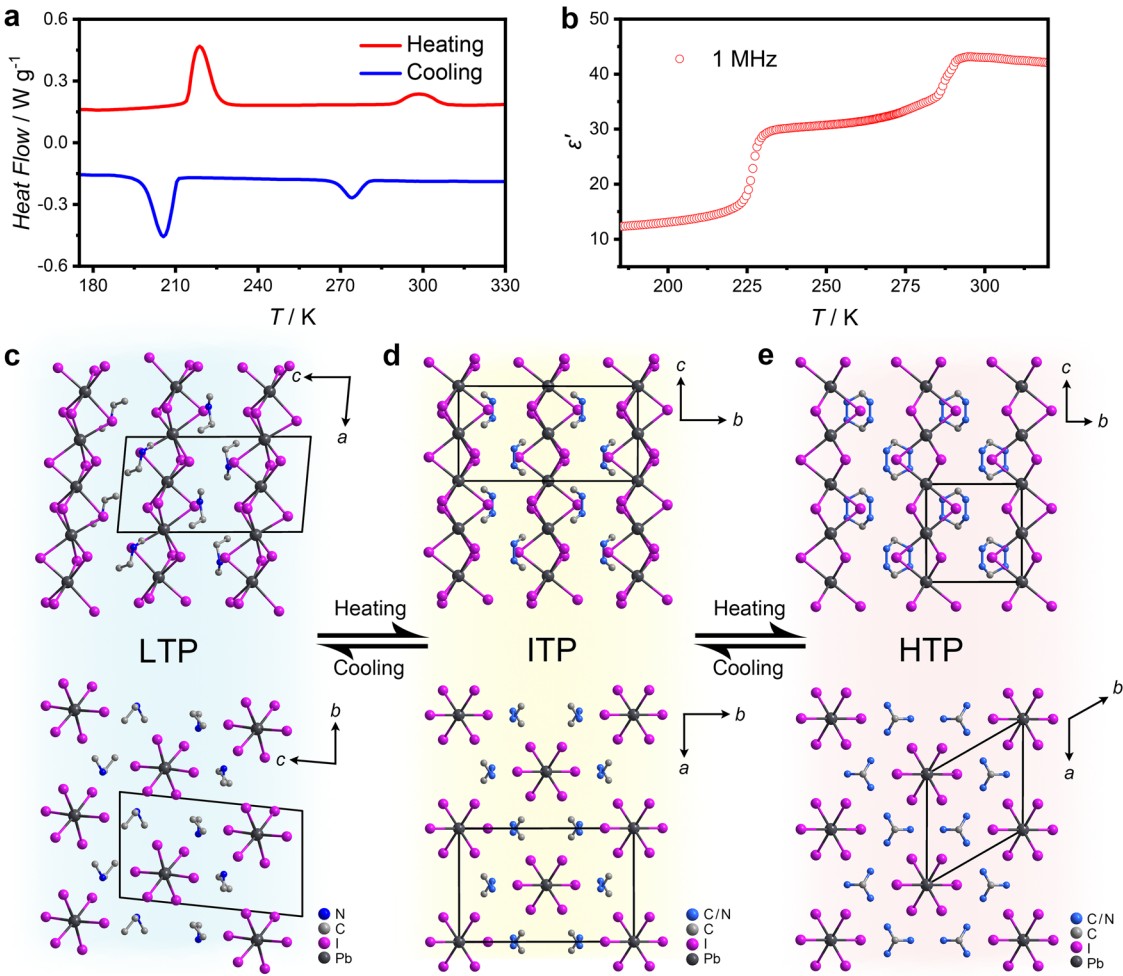

**Fig. 2 | Phase transitions of 1. a** DSC and **b** temperature-dependent dielectric constant curves of **1. c–e** Side and top views of crystal packing along Pb–I 1D chains in the LTP, ITP, and HTP. LTP, ITP and HTP represents low/intermediate/high temperature phase, respectively. Hydrogen atoms are omitted for clarity. Source data are provided as a Source Data file.

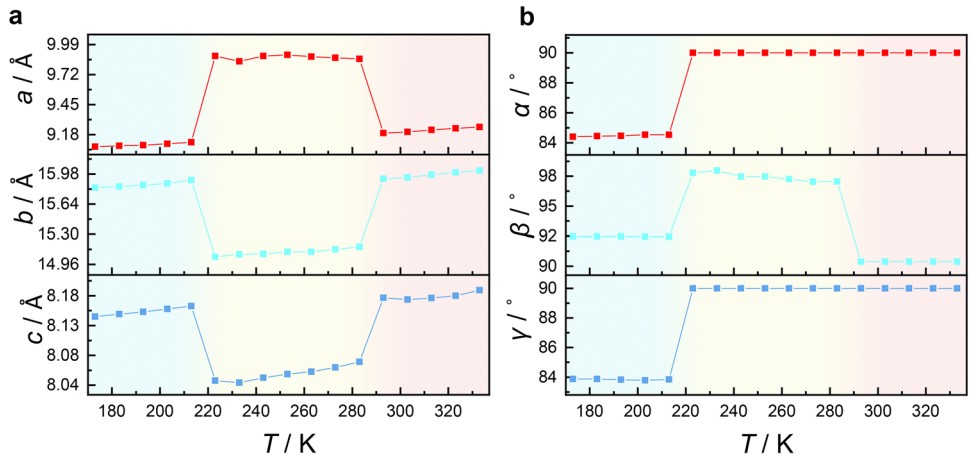

**Fig. 3 | Temperature-dependent unit-cell parameters of 1. a** Normalized unit-cell length and **b** unit-cell angle of **1**. The sharp changes in cell parameters correspond to the phase transitions. Source data are provided as a Source Data file.

(Fig. 3). Both the two transformations show features of concerted, displacive, and rapid structure phase transitions, accompanied by remarkable structural rearrangements. These characteristics, together with temperature hysteresis exhibited in DSC curves, suggest the first-order nature of the martensitic transitions.

**Ferroelastic properties**

Since the martensitic transformation is usually accompanied by the emergence of ferroelasticity, we refer to Aizu's notation[33] and confirm that both the point group changes ($2/m\mathrm{F}\bar{1}$ and $6/mmm\mathrm{F}2/m(s)$) belong to the ferroelastic types. The characteristic features

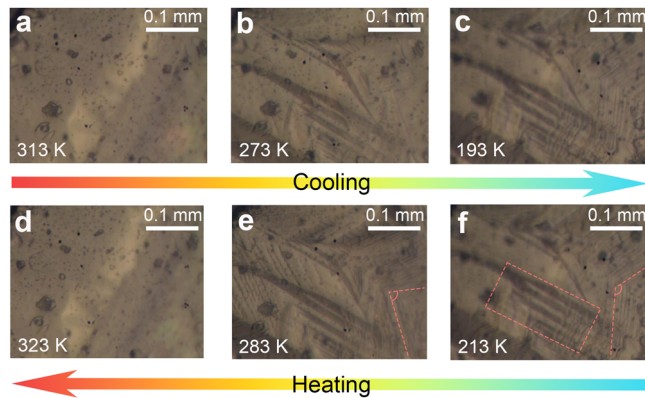

**Fig. 4 | Temperature-dependent evolution of the ferroelastic domain structures of 1. a–c** Emergence and evolution of ferroelastic domains during cooling. **d–f** Evolution and disappearance of ferroelastic domains during heating.

of ferroelastic transitions are the domain structures and hysteretic loop of strain upon stress[34,35]. The ferroelastic domain is a region in the ferroelastic phase with the same spontaneous strain orientation state. To confirm the ferroelastic transitions in **1**, polarization microscopy was used to observe the domain structure of the as-grown crystals (Fig. 4, Supplementary Movie 1). Upon cooling, the domain patterns initially occur at the HTP-ITP transition point and striped light and dark bands clearly appear on the surface of the crystals, corresponding to different orientation states (Fig. 4a, b). Keep cooling to the ITP-LTP transition point, the existing domain further evolves into a new pattern (Fig. 4c). The angle of the domain wall in the LTP is obviously smaller than that in the ITP (Fig. 4e, f). These domains observed in the cooling process completely disappear when heating to the HTP, namely, the paraelastic phase (Fig. 4d). Furthermore, the mechanically induced crystal twinning was also confirmed in the ferroelastic ITP (Supplementary Movie 2).

For ferroelastics, the spontaneous strain $x_s$ is a crucial parameter for evaluating the ferroelastic properties and it can be calculated based on the crystal lattice parameters (Supplementary Table 3)[36,37]. The associated expressions for spontaneous strain calculation are given in Supplementary Note 1. For the ITP-HTP transition belonging to 6/$mmm$F2/$m(s)$, the point group transforms from $D_{6h}$ to $C_{2h}$, accompanied by a drastic decrease of symmetry from 24 ($E$, $i$, $2C_6$, $2C_3$, $C_2$, $3C_2'$, $3C_2''$, $2S_3$, $2S_6$, $\sigma_h$, $3\sigma_d$, $3\sigma_v$) to 4 ($E$, $i$, $\sigma_h$, $C_2$) symmetric elements. The amount of possible orientation states in the ferroelastic is related to the number of symmetric elements of point groups in the paraelastic phase and ferroelastic phase. For example, six possible orientation states for the 6/$mmm$F2/$m(s)$ phase transition could be obtained from the HTP to ITP (24/4 = 6). The $x_s$ in the ITP was estimated as 0.1386 according to the following expression:

$$x_s{}^2 = \sum_{i=1}^{3}\sum_{j=1}^{3} x_{sij}{}^2 = 2w^2 + 2x_{13}{}^2 \tag{1}$$

where $w = \frac{1}{2}(x_{22} - x_{11})$.

As to the LTP-ITP transition (2/$m$F$\bar{1}$), the symmetry elements decrease from 4 ($\sigma_h$, $C_2$, $E$, $i$) to 2 ($E$, $i$) with the point group transforming from $C_{2h}$ to $C_i$. The $x_s$ was estimated as 0.1059 using the corresponding equation:

$$x_s{}^2 = \sum_{i=1}^{3}\sum_{j=1}^{3} x_{sij}{}^2 = 2x_{12}{}^2 + 2x_{23}{}^2 \tag{2}$$

The magnitude of the spontaneous strain for **1** is comparable to other ferroelastics with large spontaneous strains[31,38]. The two

ferroelastic phase transitions in **1** undergo large symmetry decreases, and the normalized unit cells before and after the transition point also show large differences which finally contribute to the large magnitude of the $x_s$ for the two ferroelastic phase transitions.

## Actuation behavior

The remarkable Pb−I chain shearing and abrupt unit cell expansion/contraction in the phase transition processes confirm the martensitic thermoelastic behaviors in **1** (Fig. 5a–c). A typical diffusionless phase transition initiates at a specific point and propagates fast throughout the whole crystal along the $c$-axis accompanied by a macroscopic shape change (Supplementary Fig. 7, Supplementary Movies 3–5).

A more intriguing phenomenon was observed for the crystal with surface defects when cooling to 280 K. The crystal suddenly bends with an angle of 169.5° and still maintains a good crystallinity (Fig. 5e, g). To our knowledge, remarkable relative slip between the [PbI₃]ₙ chains in the HTP-ITP transition is the essential condition for crystal bending (Fig. 5b, c). Furthermore, the defects on the single crystal reduce the energy required for the phase transition so that it first initiates at which the defects are. Crystal images in the ITP clearly show the interface where the shearing movement occurs, and the phase front quickly sweeps through the crystal from both sides of the interface, resulting in the crystal bending behavior (Fig. 5e). A microscopic mechanism is proposed that a twin boundary occurs around the defect which is just the place of the bending point (Fig. 5f). The bending angle estimated from the single-crystal structure analysis is about 164.5°, fairly consistent with the observed value.

As for the ITP-LTP transition, another shape change was observed at about 200 K with a bending angle slightly decreasing to 171.2°. Although the inclination angle of the Pb-I chains displays a comparable change from the ITP to LTP (Fig. 5a, b), the crystal bending angle does not present an obvious shearing change which may attribute to a multi-domain state canceling out the macroscopic changes. To clarify the microscopic changes caused by the phase transitions, the cell edges after normalization for the three phases (Table 1) are drawn in a unified coordinate system (Fig. 5d). The reversible shape change recurrence in the heating process to room temperature corresponds to the reversible phase transition characterized above. This evidence supports the temperature-initiated dual shape transformation without degradation. Such significant anisotropic contraction/expansion within a small time and temperature scale is rare in molecular crystals[19].

It needs to point out that the transition temperature ($T_{tr}$) is a key parameter for reversibly thermosalient crystals. For instance, most application scenarios in smart devices for daily use based on these materials need the transition to occur around room temperature. Other crystals, which actuate above 400 K[39] or lower than 200 K[18,40], have an extreme application scenario. As to **1**, the ITP-HTP transition is around 286 K, a temperature very close to room temperature which is well suited for ambient temperature applications (Fig. 6a). Using the reversible actuation behavior of **1**, we proposed a temperature-controlled switch (Fig. 6b). By using the shearing strain of the crystal around the near-room-temperature martensitic transition, the lead in the circuit can be connected or disconnected with the contact point. A prototype device was realized to reversibly and repeatedly switch a LED on/off via a thermal actuation (Fig. 6c, Supplementary Movie 6), demonstrating a good performance as a near-room-temperature switch. This simple device shows one of the potential applications of martensitic transformations and related mechanical changes.

## Semiconductor-related properties

Crystal **1** is supposed to be a 1D semiconductor due to the well-defined [PbI₃]ₙ chains. UV-vis absorption spectrum was first measured to show an optical cut-off at 450 nm with a corresponding bandgap of 2.3 eV deduced from the Tauc plot (Fig. 7a). DFT-PBE band structure calculation indicates an indirect bandgap

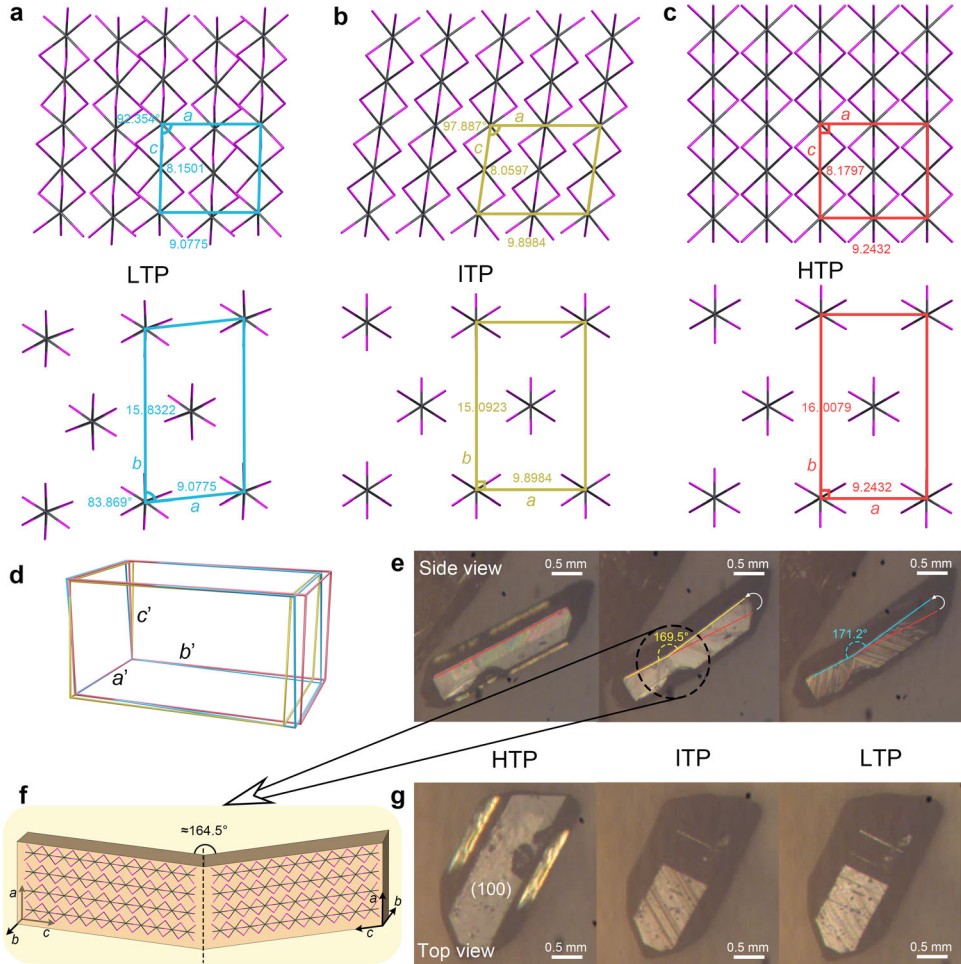

**Fig. 5 | Microscopic and macroscopic evolution for single crystals shearing. a–c** Cell shearing evolution of the martensitic phase transitions. Organic cations are omitted for clarity. **d** Comparison of the cell lattices in the LTP (cyan), ITP (yellow), and HTP (red). The axes are normalized according to the ITP. **e** Side view and **g** top view of crystal morphology during martensitic phase transitions. The HTP-ITP transition shows distinct crystal bending behavior. **f** Mechanism for the crystal bending in the ITP.

feature, showing the conduction band minimum (CBM) and the valence band maximum (VBM) at Γ and T point, respectively (Fig. 7b). The calculated bandgap value of 3.1 eV is larger than the experimental result, similar to other DFT-PBE calculations without spin-orbit coupling[41,42]. Furthermore, the partial density of states (PDOS) suggests that the CBM originates from the Pb 6p and I 5p orbits and the VBM relates to the I 5p orbit (Fig. 7c). Above analyses reveal that the inorganic $[PbI_3]_n$ contributes tremendously to the semiconducting properties.

To achieve deeper insight into the electronic properties of the compound, the electronic bands are illustrated in the partial charge density distributions of the CBM and VBM (Fig. 7d). The

former is mainly derived from the Pb 6p and I 5p orbitals while the latter is mainly made up of the I 5p orbital, agreeing with the DOS results. Both the VBM and CBM components are mainly localized at the inorganic chains along the c-axis, facilitating the generation and transportation of photo-induced charge carriers. The organic cation makes negligible contributions to the band edges except for charge balance. These results indicate that **1** should exhibit 1D electronic properties, consistent with the structural connectivity of the inorganic chains. Resistivity measurement further confirms the anisotropic electrical transport with a parallel/perpendicular ratio of 2 at 300 K with respect to the chain, obeying the 1D characteristic (Supplementary Fig. 8).

Figure 7e shows the current-voltage (I–V) curves at 300 K along the direction of the chain. The dark current and photocurrent were measured as 0.34 nA and 4 nA, respectively, with a power density of 130 mW cm⁻² at 5 V, verifying the photoconductive characteristics of **1**.

The striking structural differences brought about by martensitic transformation are supposed to have a distinct impact on carrier transport. Thus, **1** can be used as a model compound to study the relationship between the structure transition and carrier transport in 1D perovskites, which could be further explored as temperature- and phase-controlled mechano-photo-electronic switches. We measured the variable-temperature photoresponse during the HTP-ITP transition near room temperature (Fig. 7f). There is a strikingly different photoconductive response during

**Table 1 | Normalized unit cell parameters of 1 in three phases based on the ITP**

|  | LTP-173K | ITP-253K | HTP-333K |
|---|---|---|---|
| a/Å | 9.0775(2) | 9.8984(9) | 9.2432(5) |
| b/Å | 15.8322(3) | 15.0923(10) | 16.0097(9) |
| c/Å | 8.1501(3) | 8.0597(5) | 8.1797(4) |
| α/° | 84.399(2) | 90 | 90 |
| β/° | 92.354(2) | 97.887(7) | 90 |
| γ/° | 83.8690(10) | 90 | 90 |
| V/Å³ | 1158.49(5) | 1192.65(15) | 1210.44(7) |

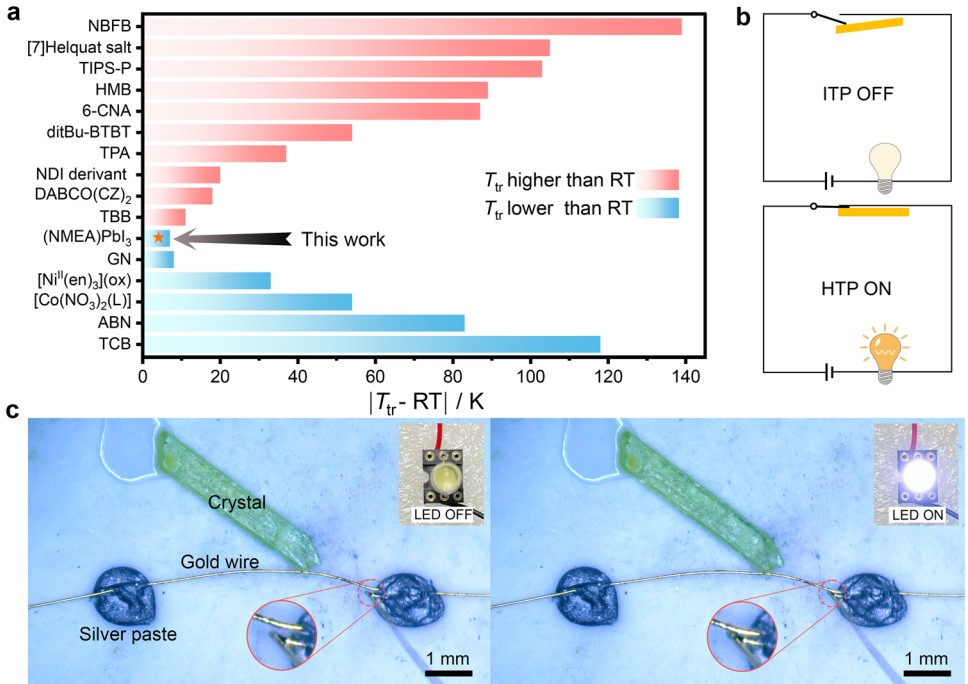

**Fig. 6 | Comparison of working temperatures of reported systems and actuation device of 1. a** Response temperature deviating from room temperature of reported thermosalient molecular crystals. NBFB[39], [7]Helquat salt[47], TIP-P[19], HMB[48], 6-CAN[49], ditBu-BTBT[19], TPA[17], NDI derivant[50], DABCO(CZ)$_2$[51], TBB[52], GN[53], [Ni$^{II}$(en)$_3$](ox)[54], [Co(NO$_3$)$_2$(L)][55], ABN[40], TCB[18]. $T_{tr}$ and RT represent phase transition temperature and room temperature of 293 K, respectively. **b** Schematic diagram and **c** demonstration of a temperature-controlled mechanical switch utilizing the reversible actuation between the ITP shearing state (OFF) and HTP non-shearing state (ON). Source data are provided as a Source Data file.

the ITP-HTP transition at 10 V bias upon a white light with an intensity of 80 mW cm$^{-2}$. From 295 to 285 K, the photocurrent shows a sudden jump from about 2.0–4.0 nA. We assume that it originates from the changes in carrier transportation caused by the cooperative shearing movement of the 1D Pb−I chains during the martensitic transformation. The optoelectronic responses in the HTP and ITP are stable and repeatable, supporting the complete occurrence of the phase transition. The sudden jumping points around 283 K may arise from the intermittent movement of the phase front during the martensitic transformation.

In summary, a 1D perovskite martensite semiconductor exhibiting ferroelastic and thermoelastic behavior has been synthesized and characterized. A large shearing of the [PbI$_3$]$_n$ chain with abrupt anisotropic expansion/contraction mainly contributes to the occurrence of martensitic transformation. The unit cell and symmetry changes correspond to a large spontaneous polarization of 0.1386. Remarkable crystal shearing during the phase transitions endows the crystal with intriguing mechanical actuation characteristics. More fascinating, the response temperature for the shearing actuation is very close to the room temperature range compared with other thermosalient crystals, which means bright application prospects. This work reveals the unique chain-like structure in 1D perovskite is prone to take relative shear motion and provides effective design ideas for martensitic actuators based on 1D structures.

## Methods
### Materials
All the analytical grade chemicals were used as received without further purification. Stoichiometric amounts of *N*-methylethylamine and lead iodide were carefully dissolved in concentrated hydroiodic acid. A clear solution was obtained after continuous stirring for 1 h at 373 K. Needle-like yellow crystals were obtained after evaporation of the solution at room temperature in a well-ventilated place for days.

### General characterizations
Powder X-ray diffraction patterns were obtained on a Rigaku SmartLab X-ray diffractometer at room temperature. The diffraction patterns were collected in the 2$\theta$ range of 5–50° with a step size of 0.02°. DSC measurements were performed on a NETZSCH DSC 200F3 instrument under N$_2$ at atmospheric pressure with a heating/cooling rate of 20 K min$^{-1}$. Ultraviolet–vis (UV–vis) absorption spectra were measured with Shimadzu UV-2600 equipped with ISR-2600Plus integrating sphere.

### Single-crystal X-ray diffraction
Crystallographic data of the compound were collected on a Rigaku Oxford Diffraction Supernova Dual Source, Cu at Zero equipped with an AtlasS2 CCD using Mo Kα radiation and an XtaLAB Synergy R, DW system, HyPix diffractometer. Rigaku CrysAlisPro software was used to collect data, refine cell, and to reduce data. SHELXL-2018 with the OLEX2 interface was used to solve the structures by direct methods[43,44]. All non-hydrogen atoms were refined anisotropically. The positions of hydrogen atoms were generated geometrically.

### Dielectric spectra
Temperature-dependent dielectric constant spectra were measured on powdered samples by using a Tonghui TH2828A impedance analyzer at frequencies of 1–1000 kHz with an applied electric field of 1 V.

### *I-V* measurement
A PDA FS380 Source meter was used under AM 1.5 G illumination. The simulated irradiation was produced by a CEL-S500/350 solar simulator (Ceaulight, 350 W) with an AM 1.5 filter. The light intensity was calibrated by a Si photodetector (CEL-NP2000-2A, Ceaulight).

### Device fabrication
The dimensions of the crystal used to assemble the photodetector device were 1×1×3 mm$^3$. Electrodes were prepared by coating silver paste on the surface of the single crystal along the *c*-axis (HTP) and

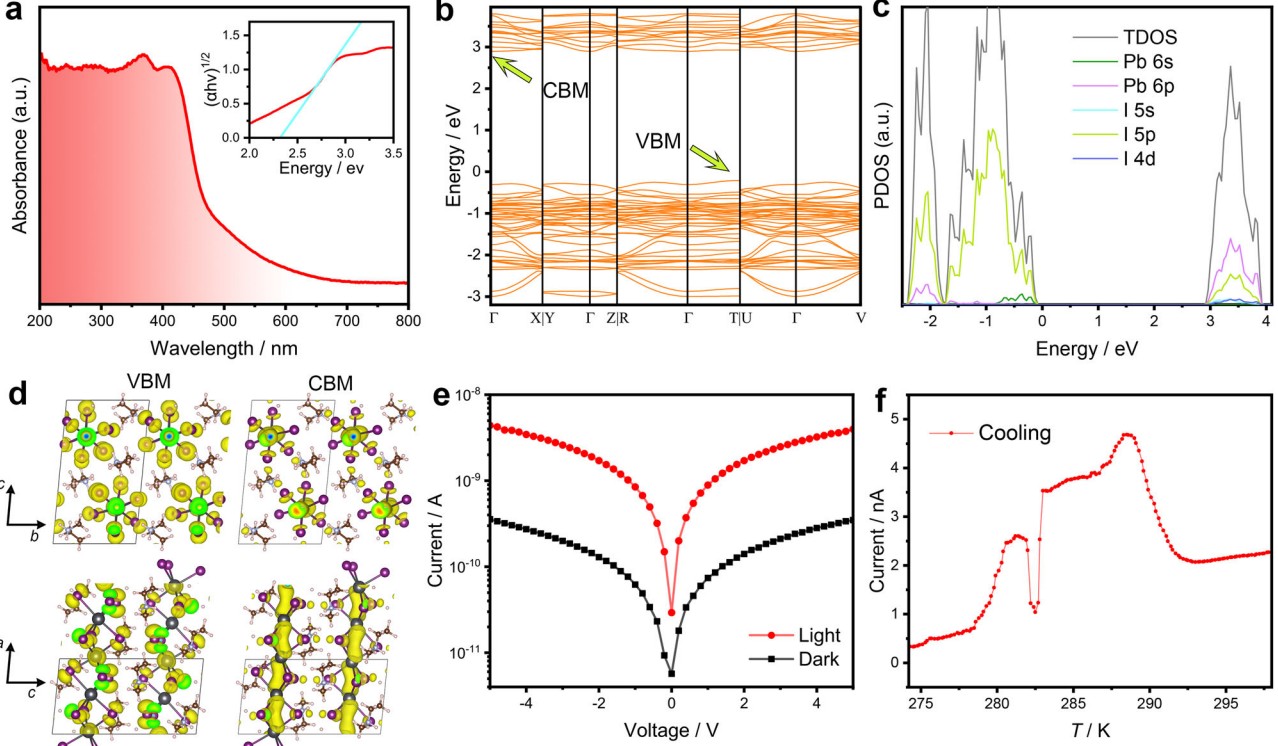

**Fig. 7 | Semiconductor-related properties of 1. a** UV−vis absorption spectrum. Inset: Fitted bandgap from the absorption spectrum. **b** DFT-PBE band structures without SOC and **c** DOS. **d** Partial charge density of the VBM and CBM. Yellow isosurface is electron cloud distribution. **e** I–V curves of the photodetector upon a white light with the intensity of 130 mW cm⁻². **f** I–T curve corresponding to the ITP-HTP transition measured at 10 V upon a white light with the intensity of 80 mW cm⁻². Source data are provided as a Source Data file.

dried at 298 K for 1 day. The effective area of the single crystal device was determined to be 0.125 mm² using a high-definition digital camera.

## DFT calculation
DFT calculations were conducted by using the VASP code. The projector augmented wave method and the Perdew−Burke−Ernzerhof (PBE) functional within the generalized gradient approximation were adopted to define the ion−electron interactions and the exchange-correlation interaction, respectively. Van der Waals interactions were evaluated by employing Grimme's dispersion-corrected semi-empirical DFT-D3 method[45]. The energy cutoff was set to 500 eV and a 4 × 4 × 2 Gamma centered grid of k-points was used. Post-processing analysis was performed by using VASPKIT[46].

## Data availability
All data generated and analyzed during this study are included in the article and its supplementary information, and are also available from corresponding authors upon request. The crystal structures generated in this study have been deposited in the Cambridge Crystallographic Data Centre under accession code CCDC: 2178487-2178489, and can be obtained free of charge from the CCDC via https://www.ccdc.cam.ac.uk/structures/. Source data are provided in this paper.

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

## Acknowledgements

This work was financially supported by the National Natural Science Foundation of China (21991144 and 21875035). We thank the Big Data Center of Southeast University for providing the facility support on the numerical calculations. B.-D.L. thanks Qing Wang for supporting the spontaneous strain calculations.

## Author contributions

B.-D.L. synthesized the sample, carried out the experiments, and wrote the original manuscript. C.-C.F. and X.-B.H. built the photoelectric test platform and carried out the semiconductor-related characterizations. C.-D.L. solved and refined the crystal structures. C.-Y.C. contributed to data analysis. W.Z. supervised the study, carried out the theoretical calculations, and revised the manuscript. All authors discussed the results and commented on the manuscript.

## Competing interests

The authors declare no competing interests.
