## [Peer Review File · Nature Communications]

Near-room-temperature martensitic actuation profited from one-dimensional hybrid perovskite structureREVIEWER COMMENTS

Reviewer #1 (Remarks to the Author):

The authors synthesized and characterized an interesting martensitic actuator based on a 1D perovskite single crystal (NMEA)PbI₃ that could be triggered around room temperature. Single crystal structure analysis reveals the mechanism that weak interaction around robust PbI chains facilitates the shearing movement, which finally initiates the actuation behavior. Furthermore, distinct ferroelasticity was verified through the reversible evolution of ferroelastic domain structure and the large spontaneous strain was achieved due to the large symmetry decrease and unit cell change after the phase transitions.

To the best of my knowledge, examples of molecular martensite with remarkable thermosensitive behavior at room temperature are very limited, and corresponding design and prediction strategies are lacking in prior work. In this manuscript, utilizing intrinsic structural characteristics of 1D perovskite to realize a martensitic actuator is instructive, which may provide a helpful strategy to obtain multifunctional martensite with stimuli-responsive properties. Besides, I learned that martensitic phase transitions are rarely discussed by researchers in the ferroelasticity field, and the authors realized martensitic and ferroelastic phase transitions in one system. This work could inspire researchers who focused on ferroelasticity and martensite to review their work and pay attention to those interesting properties like thermoelasticity, superelasticity, shape memory effect, etc., in their systems. The topic is interesting, and the results are well presented. Therefore, I recommend the publication in Nat. Commun. after minor revisions.

1. Introduction section, "The concept of martensitic transformation was first proposed in metallurgy. By quenching carbon steel in the austenite phase, martensite with improved strength, hardness, and toughness can afford excellent performance.", definition and characteristics of martensitic transformation are missing. Please supply the related information.
2. Scheme 1 points out the martensitic transformation design strategy in the 1D perovskite structure. What about other low-dimensional perovskites like 0D and 2D? It seems feasible in other low-dimensional perovskite structures.
3. According to the authors' discussion, the actuation behavior in (NMEA)PbI₃ is attributed to martensitic transition, and there is a clear microscopic correspondence. What about ferroelasticity? Is there an exact relationship between martensitic transition and ferroelasticity?
4. Fig. 6c. Check CBM contribution of I 3d orbit.
5. In CIFs of 1, "absorption correction" is not given. Did the authors forget to enter the type of absorption correction or to perform an absorption correction?

Reviewer #2 (Remarks to the Author):

This is an interesting paper that describes the thermo- and ferroelasticity of an organic-inorganic hybrid perovskite structure. The authors newly discovered near-room-temperature thermo- and ferroelasticity of (NMEA)PbI₃, a 1D type perovskite. The structural attributes of transformations are thoroughly investigated based on appropriate data sets. However, my biggest concern is the novelty of this study; martensitic transformations from perovskite structures are no longer so new to the related research fields. In order to make this work more appealing to readers, I believe that at least some of these points should be addressed in the manuscript: (1) the material designing perspectives of the authors for achieving room-temperature martensitic transition in perovskite materials, (2) a demonstration highlighting the application potential of such materials, or (3) a demonstration combining excellent optoelectronic properties of perovskites with the martensitic transformability. The authors appear to be aware of the issues, but these points should be developed further. For this reason, it is considered that the paper is not currently suitable for publication in Nature Communications.

Aside from the major concerns mentioned above, some minor ones are given below.

1. Given that ferroelasticity allows mechanical stress to convert between orientation states, it is recommended to present mechanically induced reversible twinning behavior of (NMEA)PbI₃.
2. It is necessary to assign the structure of the domains that are acquired through ferroelastic transition.
3. Methods are not fully provided in the manuscript, e.g. device fabrication methods for photocurrent measurement (Fig. 6d) and calculation methods (Fig. 6b, c)
4. The thermoelastic transformations are said to be non-displacive in the 3rd paragraph of page 5, why?

Reviewer #3 (Remarks to the Author):

Authors reported the 1D hybrid perovskite with a close-to-RT martensitic transformation characteristic. This unique chain structure endows the (NMEA)PbI₃ molecular crystal with the interesting ferroelastic and thermoelastic properties, and favors the realization of mechanical actuators. The whole manuscript is well organized and deserves to be published in this journal until the following issues are adequately addressed.

1. The title is not that accurate. The actual temperature of structural phase transition should be close to RT.
2. The shearing change is not very obvious from ITP to LTP. Can such a change be used in high-performance actuators?
3. Temperature-dependent resistivity measurement is needed to examine whether the electrical transport obeys 1D characteristic. This can directly clarify the way of electron hopping along [PbI₃]_n chain and further deepen the discussion about weak interchain and strong intrachain interactions.
4. For the last part (photoconductive measurement), the wavelength of excitation source is not given. I think the optoelectronic behavior after martensitic transformation is very crucial to demonstrate the complete occurrence of phase transition for comparison. This part of work needs some supplementary information.
5. I am not sure whether the reference number can be inserted in the caption section for the Fig. 5. In general, they are often marked in pictures.
6. For the panel b in Fig. 5, some necessary demonstration should be supplied to explain the reversible mechanism.

Reviewer #1 (Remarks to the Author):

The authors synthesized and characterized an interesting martensitic actuator based on a 1D perovskite single crystal (NMEA)PbI₃ that could be triggered around room temperature. Single crystal structure analysis reveals the mechanism that weak interaction around robust PbI chains facilitates the shearing movement, which finally initiates the actuation behavior. Furthermore, distinct ferroelasticity was verified through the reversible evolution of ferroelastic domain structure and the large spontaneous strain was achieved due to the large symmetry decrease and unit cell change after the phase transitions.

To the best of my knowledge, examples of molecular martensite with remarkable thermosensitive behavior at room temperature are very limited, and corresponding design and prediction strategies are lacking in prior work. In this manuscript, utilizing intrinsic structural characteristics of 1D perovskite to realize a martensitic actuator is instructive, which may provide a helpful strategy to obtain multifunctional martensite with stimuli-responsive properties. Besides, I learned that martensitic phase transitions are rarely discussed by researchers in the ferroelasticity field, and the authors realized martensitic and ferroelastic phase transitions in one system. This work could inspire researchers who focused on ferroelasticity and martensite to review their work and pay attention to those interesting properties like thermoelasticity, superelasticity, shape memory effect, etc., in their systems.

The topic is interesting, and the results are well presented. Therefore, I recommend the publication in Nat. Commun. after minor revisions.

Response:

Thanks!

1. Introduction section, “The concept of martensitic transformation was first proposed in metallurgy. By quenching carbon steel in the austenite phase, martensite with

improved strength, hardness, and toughness can afford excellent performance.”, definition and characteristics of martensitic transformation are missing. Please supply the related information.

Response:

We added a definition of martensitic transformation in the Introduction section:

Martensitic transformation is a displacive-type solid-state phase transition with basic characteristics of diffusionlessness, first order, low transition barrier, ultrafast kinetics, cooperative displacement, habit plane, and structural reversibility.

2. Scheme 1 points out the martensitic transformation design strategy in the 1D perovskite structure. What about other low-dimensional perovskites like 0D and 2D? It seems feasible in other low-dimensional perovskite structures.

Response:

We had considered this point when preparing the manuscript. There are two opposite trends in martensitic transformation and semiconduction. The lower the dimensionality, the easier the martensitic transformation but the poorer the semiconduction.

In 2D perovskite structures, relative slipping of the infinite 2D inorganic grids is largely prohibited due to almost unsurmountable energy barriers. But they are good candidates as semiconductors. In 0D structures, the organic and inorganic components may favor relative slipping, but the 0D electronic structures are undesired due to poor semiconducting properties that hinder the prospect of multifunctional applications.

In contrast, the 1D perovskite structures have well-balanced crystallographic and electronic dimensionalities, which allow the coexistence of martensitic transformation and semiconduction. Therefore, we think that the 1D perovskite structures are the most ideal platform for the design and screening of candidates for martensitic transformation in semiconducting materials.

In addition, although some 1D perovskite structures (*Chem. Mater.* **34**, 3518-3524 (2022); *Inorg. Chem.* **61**, 2219-2226 (2022); *J. Phys. Chem. Lett.* **12**, 5221-5227 (2022);

Chem. Commun. **57**, 6292-6295 (2022); *Inorg. Chem.* **56**, 4918-4927 (2017)) were reported to show ferroelasticity, their nature of martensitic transformations has been unidentified by the authors, not to mention the combined actuation and semiconduction properties.

3. According to the authors' discussion, the actuation behavior in (NMEA)PbI₃ is attributed to martensitic transition, and there is a clear microscopic correspondence. What about ferroelasticity? Is there an exact relationship between martensitic transition and ferroelasticity?

Response:

The basic characteristics of martensitic transition are the diffusionlessness and cooperative displacement of atoms and ions during the phase transition. Accordingly, the phase transition is also classified as a displacive type. In comparison, the ferroelastic phase transition is accompanied by the spontaneous strain occurring in the ferroelastic phase which is characterized by a unique strain-stress hysteresis response. The paraelastic-ferroelastic phase transition is also a diffusionless type. Historically, the ferroelastics are treated as mechanical counterparts of ferromagnets and ferroelectrics. The ferroelastic phase transition obeys certain symmetry-breaking rules derived by Aizu. Therefore, the martensitic and ferroelastic transitions are not equivalent according to their own definitions.

However, both of the transitions share some common features. In most cases, the ferroelastic one can be treated as a subgroup of the martensitic one with the same microscopic mechanism. For example, it is said that "In addition to mechanically induced polymorph transitions, mechanically induced reversible twinning is another type of martensitic transition known as ferroelasticity" (Y. Diao, et al. Martensitic transition in molecular crystals for dynamic functional materials. *Chem. Soc. Rev.* **49**, 8287-8314 (2020)). For ferroelastic phase transitions, potential mechanical-related properties with martensitic transformations need to be emphasized.

Our study represents a suitable example to illustrate the close connection between the

martensitic and ferroelastic transitions and affords a new perspective on the two well-known topics. To explain the relationship between the martensitic and ferroelastic phase transitions, we added a description in the Introduction section as below.

It should be mentioned that the ferroelastic phase transition is often thought to be a subgroup of martensitic transformation. Although they share the basic characteristics, the former is characterized by spontaneous strain and a unique strain-stress hysteresis response in the ferroelastic phase which obeys certain symmetry-breaking rules when the phase transition occurs.

4. Fig. 6c. Check CBM contribution of I 3d orbit.

Response:

We checked the contribution of I orbit and revised the label in Fig 6.

5. In CIFs of 1, “absorption correction” is not given. Did the authors forget to enter the type of absorption correction or to perform an absorption correction?

Response:

“MULTI-SCAN” was used in absorption correction in single-crystal diffraction. We revised the CIFs and updated the data in CCDC.

Reviewer #2 (Remarks to the Author):

This is an interesting paper that describes the thermo- and ferroelasticity of an organic-inorganic hybrid perovskite structure. The authors newly discovered near-room-temperature thermo- and ferroelasticity of (NMEA)PbI₃, a 1D type perovskite. The structural attributes of transformations are thoroughly investigated based on appropriate data sets. However, my biggest concern is the novelty of this study; martensitic transformations from perovskite structures are no longer so new to the related research fields.

Response:

We thank the Reviewer for the insightful comment.

The study of martensites and transformations has a long history due to their important applications. Recently, much interest has been aroused in molecule-based martensites with promising potential in soft devices. However, examples of hybrid organic-inorganic perovskites that clearly point out to be martensites have been rarely reported. Although many hybrid organic-inorganic perovskites have been reported to be ferroelastic, the authors did not reveal their nature of the martensitic transition, let alone mention the study of other martensite-related mechanical properties. See examples: *Chem. Mater.* **34**, 3518-3524 (2022); *Inorg. Chem.* **61**, 2219-2226 (2022); *J. Phys. Chem. Lett.* **12**, 5221-5227 (2022); *Chem. Commun.* **57**, 6292-6295 (2022); *Inorg. Chem.* **56**, 4918-4927 (2017). After careful structural analysis, we conclude that these systems are actually also martensites. However, this insightful point has been never recognized and clearly pointed out by the authors. Our work disclosed the common features of the two types of phase transitions and represents a good example to illustrate the close connection between the martensitic and ferroelastic transitions. More importantly, the combination of electronic, optical, and mechanical properties of the hybrid organic-inorganic perovskites opens a way for novel multifunctional materials. Besides, our work proposes a strategy for constructing martensites in 1D perovskites. It is instructive and practicable for achieving room-temperature martensitic transition

in perovskite materials and helpful in realizing multifunctional stimuli-responsive materials based on 1D perovskite martensite (*See page 6-7 for details*).

What's more, by following the Reviewer's constructive suggestions, the demonstration of using the actuation property for the mechanical switch was successfully achieved. This is a good demonstration of the practical application potential of such kinds of materials (*See page 8 for details*).

In order to make this work more appealing to readers, I believe that at least some of these points should be addressed in the manuscript:

(1) the material designing perspectives of the authors for achieving room-temperature martensitic transition in perovskite materials,

Response:

Thanks for this suggestion!

We would like to emphasize two points from the material designing perspective.

We first considered the impact of structural dimensions on martensitic transformations. The 1D perovskite structure is thought the most ideal structural model for the design of martensitic transformation. In 2D perovskite structures, the ionic interactions between the interdigitated organic spacers and inorganic sheets are much stronger than van der Waals forces so relative slipping is largely prohibited. In 0D structures, the almost isotropic interactions may, on the other side, reduce the possibility of relative slipping. In contrast, the 1D perovskite structures have both moderate interactions among chains and crystallographic anisotropy, which ease the martensitic transformations.

The second consideration is the tuning of the transition temperature which is a key parameter for applications. There are many practical methods to tune the transition temperatures of hybrid organic-inorganic perovskites to the room temperature range, such as group/component substitution strategy (e.g., halogen and metal), isotope effect, and doping/mixing method. That is to say, as the ideal platform for designing molecular martensite, 1D hybrid organic-inorganic perovskites possess a series of feasible

methods to adjust the phase transition temperature.

We added the following descriptions in the Introduction section to stress these points:

We hope to expand molecular martensites to hybrid halide perovskite materials for their outstanding optical and electrical properties. From the material designing perspective, the structural dimensionality has a great impact on the martensitic transitions. As a representative of low-dimensional perovskites, 1D perovskite has superior structural anisotropy features in addition to excellent stability. Unlike the isotropic Pb–I bonds in 3D perovskites such as MAPbI₃ (MA = methylammonium), the 1D perovskite structure only has strong bonds within the Pb-I chain. Due to the blocking of organic cations, the 1D chains are bound together by weak van der Waals interactions, The moderate interactions among the chains and crystallographic anisotropy can ease the interchain movement and further martensitic transformation (Scheme 1). In contrast, 2D and 0D perovskites are less favored structures which suffer from either strong interlayer interactions that impede relative slipping-like displacements or undesired electronic structures and poor semiconducting properties. Meanwhile, the transition temperatures of the perovskites, as a key parameter for practical applications, can be easily tuned to room temperature by various methods such as group/component substitution strategy (e.g., halogen and metal), isotope effect, and doping/mixing.

(2) a demonstration highlighting the application potential of such materials, or (3) a demonstration combining excellent optoelectronic properties of perovskites with the martensitic transformability. The authors appear to be aware of the issues, but these points should be developed further. For this reason, it is considered that the paper is not currently suitable for publication in Nature Communications.

Response:

Thanks for these suggestions which greatly improved our work! We tried our best to demonstrate both (2) and (3) as shown below.

Response to (2)

We realized a working prototype to switch on/off a LED by using the shearing strain of the crystal around the near-room-temperature-phase transition. This simple device shows one of the potential applications of martensitic transformations and related mechanical changes. Supplementary Movie 6 shows the prototype for reversibly switching a LED via a thermal actuation. Following descriptions were added in the main text:

Using the reversible actuation behavior of **1**, we proposed a temperature-controlled switch (Fig. 5b). By using the shearing strain of the crystal around the near-room-temperature martensitic transition, the lead in the circuit can be connected or disconnected with the contact point. A prototype device was realized to reversibly and repeatedly switch a LED on/off via a thermal actuation (Supplementary Movie 6), demonstrating a good performance as a near-room-temperature switch. This simple device shows one of the potential applications of martensitic transformations and related mechanical changes.

Fig. 5c Demonstration of a temperature-controlled mechanical switch by means of the reversible actuation between the ITP shearing state (OFF) and HTP non-shearing state (ON).

Response to (3)

We set up a homemade instrument to measure the variable-temperature photoresponse during the HTP-ITP transition near the room temperature. The results are consistent with our original expectation that there is a strikingly different photoconductive response during the ITP-HTP transition. We assume that it originated from the changes of carrier transportation caused by the ordered

shearing movement of the 1D Pb-I chains during the martensitic transformation.

Following descriptions were added in the main text:

The striking structural differences brought by martensitic transformation is supposed to have a distinct impact on carrier transport. Thus, **1** can be used as a model compound to study the relationship between the structure transition and carrier transport in 1D perovskites, which could be further explored as temperature- and phase-controlled mechano-photo-electronic switches. We measured the variable-temperature photoresponse during the HTP-ITP transition near room temperature (Figure 6f). There is a strikingly different photoconductive response during the ITP-HTP transition at 10 V bias upon a white light with an intensity of 80 mW/cm². From 295 and 285 K, the photocurrent shows a sudden jump from about 2.0 to 4.0 nA. We assume that it originated from the changes in carrier transportation caused by the cooperative shearing movement of the 1D Pb-I chains during the martensitic transformation. The optoelectronic responses in the HTP and ITP are stable and repeatable, supporting the complete occurrence of the phase transition. The sudden jumping points around 283 K may arise from the intermittent movement of the phase front during the martensitic transformation.

Fig. 6f $I-T$ curve corresponding to the ITP-HTP transition measured at 10 V upon a white-light intensity of 80 mW/cm².

Aside from the major concerns mentioned above, some minor ones are given below.

1. Given that ferroelasticity allows mechanical stress to convert between orientation states, it is recommended to present mechanically induced reversible twinning behavior of (NMEA)PbI₃.

Response:

The mechanically induced reversible twinning behavior has been tested. As is shown in Supplementary Movie 2, the reversible twinning behavior of (NMEA)PbI₃ crystal is presented in the ferroelastic ITP.

2. It is necessary to assign the structure of the domains that are acquired through ferroelastic transition.

Response:

Ferroelastic crystal has several orientation states (OS). These OS are different in spontaneous strain but identical or enantiomorphous in crystal structure. Therefore, the light and dark strips in Fig. 3 are independent ferroelastic domains with the same OS. The formation of ferroelastic domains can be observed when the crystal transforms from the paraelastic to the ferroelastic phase (HTP-ITP). Due to the direction of the domain wall corresponding to the changes of point groups, the ferroelastic domain further evolves during the transition from ferroelastic to a new ferroelastic phase (ITP-LTP). See reference: J. Sapriel. Domain-wall orientations in ferroelastics. *Phys. Rev. B* **12**, 5128-5140 (1975). Following descriptions were added in the main text:

The ferroelastic domain is a region in the ferroelastic phase with the same spontaneous strain orientation state.

Upon cooling, the domain patterns initially occur at the HTP-ITP transition point and striped light and dark bands clearly appear on the surface of the crystals, corresponding to different orientation states (Fig. 3 and Supplementary Movie 1)

3. Methods are not fully provided in the manuscript, e.g. device fabrication methods for photocurrent measurement (Fig. 6d) and calculation methods (Fig. 6b, c)

Response:

These methods were added in the Methods section:

Device fabrication. The dimensions of the (NMEA)PbI₃ crystal used to assemble the photodetector device were 1×1×3 mm³. Electrodes were prepared by coating silver paste on the surface of the single crystal along the *c*-axis (HTP) and dried at 298 K for 1 day. The effective area of the single crystal device was determined to be 0.125 mm² using a high-definition digital camera.

DFT calculation. DFT calculations were conducted by using the VASP code. The projector augmented wave method and the Perdew–Burke–Ernzerhof (PBE) functional within the generalized gradient approximation were adopted to define the ion-electron

interactions and the exchange-correlation interaction, respectively. Van der Waals interactions was evaluated by employing Grimme's dispersion-corrected semi-empirical DFT-D3 method. The energy cutoff was set to 500 eV and a 4×4×2 Gamma centered grid of k-points was used. Post-processing analysis was performed by using VASPKIT.

4. The thermoelastic transformations are said to be non-displacive in the 3rd paragraph of page 5, why?

Response:

Thanks for pointing out this mistake. The thermoelastic transformations are surely displacive. The sentence was revised as:

Both the two transformations show features of concerted, displacive, and rapid structure phase transitions, accompanied by remarkable structural rearrangements.

Reviewer #3 (Remarks to the Author):

Authors reported the 1D hybrid perovskite with a close-to-RT martensitic transformation characteristic. This unique chain structure endows the (NMEA)PbI₃ molecular crystal with the interesting ferroelastic and thermoelastic properties, and favors the realization of mechanical actuators. The whole manuscript is well organized and deserves to be published in this journal until the following issues are adequately addressed.

Response:

Thanks!

1. The title is not that accurate. The actual temperature of structural phase transition should be close to RT.

Response:

Thanks for this suggestion, we have revised the title as:

Near-room-temperature martensitic actuation profited from one-dimensional hybrid perovskite structure

2. The shearing change is not very obvious from ITP to LTP. Can such a change be used in high-performance actuators?

Response:

The ITP-LTP transition only shows a tiny shearing movement. So, it is not suitable for the use as a high-performance actuator, but may have applications in conditions of low temperature and high accuracy.

In fact, we focus on the HTP-ITP transition which shows a remarkable actuation behavior near room temperature. This is the focus of our study. We believe it is the HTP-ITP transition that endows the crystal with fundamental research and potential application values.

3. Temperature-dependent resistivity measurement is needed to examine whether the electrical transport obeys 1D characteristic. This can directly clarify the way of electron hopping along $[\text{PbI}_3]_n$ chain and further deepen the discussion about weak interchain and strong intrachain interactions.

Response:

Thanks for the suggestion!

We measured the temperature-dependent resistivity of $(\text{NMEA})\text{PbI}_3$ crystal and added the result in Supplementary Information. As shown in Supplementary Fig. 8, with temperature increasing, a negative tendency of resistivity reveals that $(\text{NMEA})\text{PbI}_3$ possesses semiconductor characteristics. Moreover, resistivity parallel to the 1D chains is larger than that perpendicular to the 1D chains. The results directly clarify the way of electron hopping along $[\text{PbI}_3]_n$ chain and further deepen the discussion about the weak interchain and strong intrachain interactions.

Supplementary Fig. 8. Temperature-dependent resistivity of $(\text{NMEA})\text{PbI}_3$.

Besides, we performed partial charge density calculation for the VBM and CBM of the

(NMEA)PbI₃ crystal to help prove the charge transfer along the 1D chain direction. Figure 6d and a paragraph were added in the main text as shown below.

Figure 6. **d** Partial charge density for the VBM and CBM states of (NMEA)PbI₃.

To achieve deeper insight into the electronic properties of the compound, the electronic bands are illustrated in the partial charge density distributions of the CBM and VBM (Figure 6d). The former is mainly derived from the Pb 6p and I 5p orbitals while the latter is mainly made up of the I 5p orbital, agreeing with the DOS results. Both the VBM and CBM components are mainly localized at the inorganic chains along the *c*-axis, facilitating the generation and transportation of photo-induced charge carriers. The organic cation makes negligible contributions to the band edges except for charge balance. These results indicate that **1** should exhibit 1D electronic properties, consistent with the structural connectivity of the inorganic chains. Resistivity measurement further confirms the anisotropic electrical transport with a parallel/perpendicular ratio of 2 at 300 K with respect to the chain, obeying the 1D characteristic (Supplementary Fig. 8.).

4. For the last part (photoconductive measurement), the wavelength of excitation source

is not given. I think the optoelectronic behavior after martensitic transformation is very crucial to demonstrate the complete occurrence of phase transition for comparison. This part of work needs some supplementary information.

Response:

We used a white light as an excitation source and the information was added to the caption of Fig. 6.

We set up a homemade instrument to measure the variable-temperature photoresponse during the HTP-ITP transition near the room temperature. The results are consistent with our original expectation that there is a strikingly different photoconductive response during the ITP-HTP transition. We assume that it originated from the changes of carrier transportation caused by the ordered shearing movement of the 1D Pb-I chains during the martensitic transformation. The optoelectronic responses in the HTP and ITP are stable and repeatable, supporting the complete occurrence of the phase transition.

Fig. 6f $I-T$ curve corresponding to the ITP-HTP transition measured at 10 V upon a white light with intensity of 80 mW/cm².

5. I am not sure whether the reference number can be inserted in the caption section for the Fig. 5. In general, they are often marked in pictures.

Response:

It is OK. We referred to some recently published articles which inserted reference numbers in the caption of figures, e.g., *Nat. Commun.* **13**, 1551 (2022); *Nat. Commun.* **13**, 2293 (2022); *Nat. Commun.* **13**, 2823 (2022).

6. For the panel b in Fig. 5, some necessary demonstration should be supplied to explain the reversible mechanism.

Response:

We reorganized and supplied more details for the reversible mechanism in Fig. 5. Besides, a device demonstration is presented in Supplementary Movie 6.

We realized a working prototype to switch on/off a LED by using the shearing strain of the crystal around the near-room-temperature-phase transition (Fig. 5c). This simple device shows one of the potential applications of martensitic transformations and related mechanical changes. A more intuitive reversible mechanism can be seen in Supplementary Movie 6.

Fig. 5. b Schematic diagram and **c** demonstration of a temperature-controlled mechanical switch by means of the reversible actuation between the ITP shearing state (OFF) and HTP non-shearing state (ON).

REVIEWER COMMENTS

Reviewer #1 (Remarks to the Author):

The raised points have been well solved. According to this reviewer's judgment, the present work will be an outstanding and exciting contribution to ferroelasticity and related fields. Thus, this reviewer does recommend the publication of the present article in Nature Communications without further changes.

Reviewer #2 (Remarks to the Author):

Again, the discovery of a 1D type perovskite that represent both near room-temperature thermoelasticity and ferroelasticity is an appealing work. There were a few critical concerns, e.g., lack of material designing perspectives, and demonstration that highlights the application potential. However, in the revised manuscript, all the comments and questions are adequately addressed. Given the novelty of this work, I recommend acceptance for publication in Nature Communications.

Reviewer #3 (Remarks to the Author):

The authors have addressed all the concerns that I have raised in the previous review comment. I recommend this manuscript can be accepted in the current form.

Reviewer #1 (Remarks to the Author):

The raised points have been well solved. According to this reviewer's judgment, the present work will be an outstanding and exciting contribution to ferroelasticity and related fields. Thus, this reviewer does recommend the publication of the present article in Nature Communications without further changes.

Response:

We thank the Reviewer for spending time and effort to help increase the impact of our work! The insightful comments on ferroelasticity and martensite are of great significance to our work.

Reviewer #2 (Remarks to the Author):

Again, the discovery of a 1D type perovskite that represent both near room-temperature thermoelasticity and ferroelasticity is an appealing work. There were a few critical concerns, e.g., lack of material designing perspectives, and demonstration that highlights the application potential. However, in the revised manuscript, all the comments and questions are adequately addressed. Given the novelty of this work, I recommend acceptance for publication in Nature Communications.

Response:

We appreciate the constructive suggestions on our manuscript, including the ones to provide material designing perspectives and to demonstrate a prototype device, which greatly improved our work. Thanks again!

Reviewer #3 (Remarks to the Author):

The authors have addressed all the concerns that I have raised in the previous review comment. I recommend this manuscript can be accepted in the current form.

Response:

We would like to express our gratitude to the Reviewer for the beneficial suggestions, that is, examining the 1D electrical characteristic and testing optoelectronic behavior about martensitic transformation, which helped improve the quality of our article.